# VideoComposer: Compositional Video Synthesis with Motion Controllability

**Xiang Wang**[1*]    **Hangjie Yuan**[1*]    **Shiwei Zhang**[1*]    **Dayou Chen**[1*]    **Jiuniu Wang**[1]
**Yingya Zhang**[1]    **Yujun Shen**[2]    **Deli Zhao**[1]    **Jingren Zhou**[1]

[1]Alibaba Group    [2]Ant Group
{xiaolao.wx, yuanhangjie.yhj, zhangjin.zsw}@alibaba-inc.com
{dayou.cdy, wangjiuniu.wjn, yingya.zyy, jingren.zhou}@alibaba-inc.com
{shenyujun0302, zhaodeli}@gmail.com

## Abstract

The pursuit of controllability as a higher standard of visual content creation has yielded remarkable progress in customizable image synthesis. However, achieving controllable video synthesis remains challenging due to the large variation of temporal dynamics and the requirement of cross-frame temporal consistency. Based on the paradigm of compositional generation, this work presents `VideoComposer` that allows users to flexibly compose a video with textual conditions, spatial conditions, and more importantly temporal conditions. Specifically, considering the characteristic of video data, we introduce the motion vector from compressed videos as an explicit control signal to provide guidance regarding temporal dynamics. In addition, we develop a Spatio-Temporal Condition encoder (STC-encoder) that serves as a unified interface to effectively incorporate the spatial and temporal relations of sequential inputs, with which the model could make better use of temporal conditions and hence achieve higher inter-frame consistency. Extensive experimental results suggest that `VideoComposer` is able to control the spatial and temporal patterns simultaneously within a synthesized video in various forms, such as text description, sketch sequence, reference video, or even simply hand-crafted motions. The code and models are publicly available at https://videocomposer.github.io.

## 1 Introduction

Driven by the advances in computation, data scaling and architectural design, current visual generative models, especially diffusion-based models, have made remarkable strides in automating content creation, empowering designers to generate realistic images or videos from a textual prompt as input [24, 48, 53, 62]. These approaches typically train a powerful diffusion model [48] conditioned by text [23] on large-scale video-text and image-text datasets [2, 51], reaching unprecedented levels of fidelity and diversity. However, despite this impressive progress, a significant challenge remains in the limited controllability of the synthesis system, which impedes its practical applications.

Most existing methods typically achieve controllable generation mainly by introducing new conditions, such as segmentation maps [48, 65], inpainting masks [72] or sketches [38, 79], in addition to texts. Expanding upon this idea, Composer [28] proposes a new generative paradigm centered on the concept of *compositionality*, which is capable of composing an image with various input conditions, leading to remarkable flexibility. However, Composer primarily focuses on considering

---

*Equal contribution.

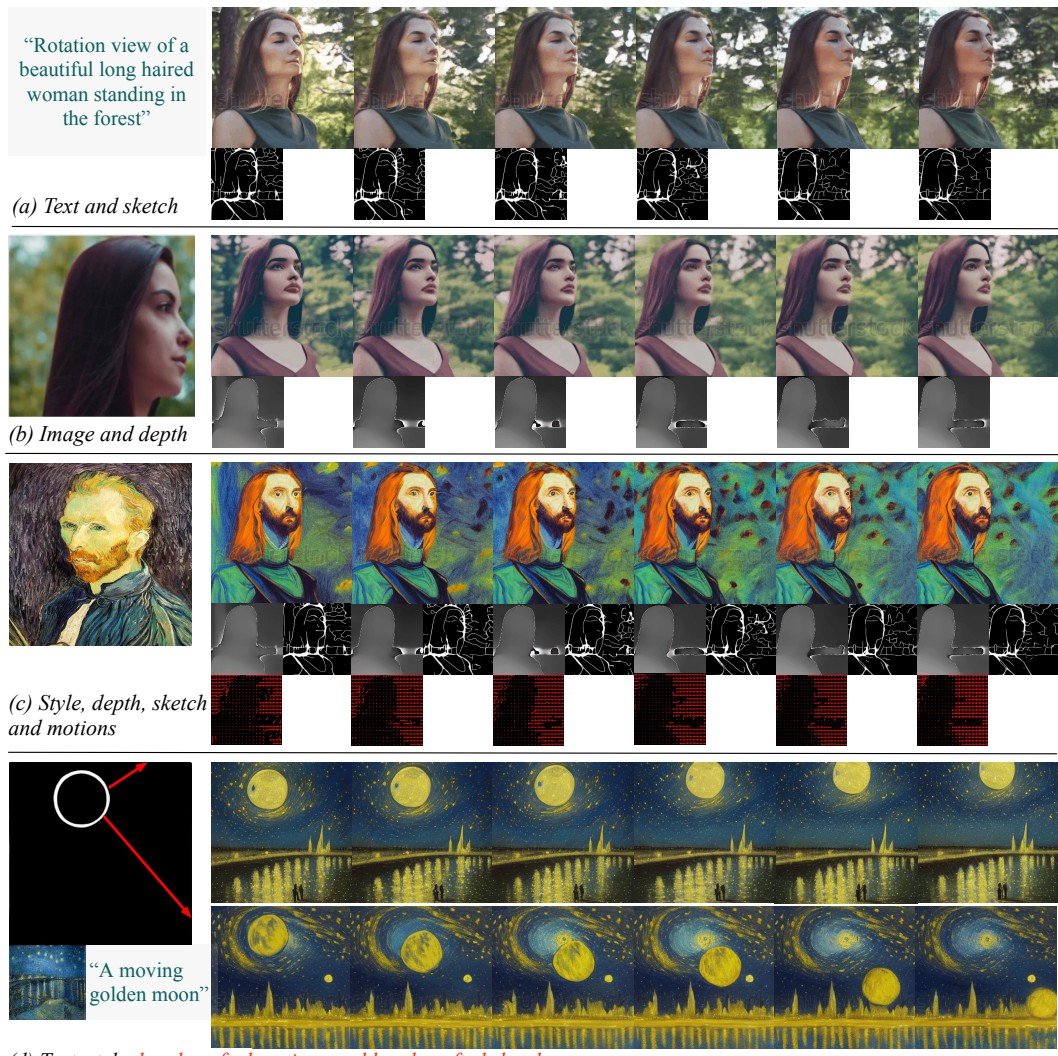

*(a) Text and sketch*

*(b) Image and depth*

*(c) Style, depth, sketch and motions*

*(d) Text, style, hand-crafted motions and hand-crafted sketch*

Figure 1: **Compositional video synthesis. (a-c)** `VideoComposer` is capable of generating videos that adhere to textual, spatial and temporal conditions or their subsets; **(d)** `VideoComposer` can synthesize videos conforming to expected motion patterns (red stroke) and shape patterns (white stroke) derived from two simple strokes.

multi-level conditions within the spatial dimension, hence it may encounter difficulties when comes to video generation due to the inherent properties of video data. This challenge arises from the complex temporal structure of videos, which exhibits a large variation of temporal dynamics while simultaneously maintaining temporal continuity among different frames. Therefore, incorporating suitable temporal conditions with spatial clues to facilitate controllable video synthesis becomes significantly essential.

Above observations motivate the proposed `VideoComposer`, which equips video synthesis with improved controllability in both spatial and temporal perception. For this purpose, we decompose a video into three kinds of representative factors, *i.e.*, textual condition, spatial conditions and the crucial temporal conditions, and then train a latent diffusion model to recompose the input video conditioned by them. In particular, we introduce the video-specific *motion vector* as a kind of temporal guidance during video synthesis to explicitly capture the inter-frame dynamics, thereby providing direct control over the internal motions. To ensure temporal consistency, we additionally present a unified STC-encoder that captures the spatio-temporal relations within sequential input utilizing cross-frame attention mechanisms, leading to an enhanced cross-frame consistency of the output videos. Moreover, STC-encoder serves as an interface that allows for efficient and unified utilization of the control signals from various condition sequences. As a result, `VideoComposer` is

capable of flexibly composing a video with diverse conditions while simultaneously maintaining the synthesis quality, as shown in Fig. 1. Notably, we can even control the motion patterns with simple hand-crafted motions, such as an arrow indicating the moon's trajectory in Fig. 1d, a feat that is nearly impossible with current methods. Finally, we demonstrate the efficacy of `VideoComposer` through extensive qualitative and quantitative results, and achieve exceptional creativity in the various downstream generative tasks.

## 2 Related work

**Image synthesis with diffusion models.** Recently, research efforts on image synthesis have shifted from utilizing GANs [18], VAEs [31], and flow models [14] to diffusion models [9, 19, 23, 32, 34, 54, 59, 74, 80, 81] due to more stable training, enhanced sample quality, and increased flexibility in a conditional generation. Regarding image generation, notable works such as DALL-E 2 [46] and GLIDE [40] employ diffusion models for text-to-image generation by conducting the diffusion process in pixel space, guided by CLIP [44] or classifier-free approaches. Imagen [50] introduces generic large language models, *i.e.*, T5 [45], improving sample fidelity. The pioneering work LDMs [48] uses an autoencoder [15] to reduce pixel-level redundancy, making LDMs computationally efficient. Regarding image editing, pix2pix-zero [42] and prompt-to-prompt editing [21] follow instructional texts by manipulating cross-attention maps. Imagic [29] interpolates between an optimized embedding and the target embedding derived from text instructions to manipulate images. DiffEdit [11] introduces automatically generated masks to assist text-driven image editing. To enable conditional synthesis with flexible input, ControlNet [79] and T2I-Adapter [38] incorporate a specific spatial condition into the model, providing more fine-grained control. One milestone, Composer [28], trains a multi-condition diffusion model that broadly expands the control space and displays remarkable results. Nonetheless, this compositionality has not yet been proven effective in video synthesis, and `VideoComposer` aims to fill this gap.

**Video synthesis with diffusion models.** Previous methods [13, 58, 67] usually adopt GANs for video synthesis. Recent research has demonstrated the potential of employing diffusion models for high-quality video synthesis [5, 20, 25, 30, 36, 62, 75]. Notably, ImagenVideo [24] and Make-A-Video [53] both model the video distribution in pixel space, which limits their applicability due to high computational demands. In contrast, MagicVideo [83] models the video distribution in the latent space, following the paradigm of LDMs [48], significantly reducing computational overhead. With the goal of editing videos guided by texts, VideoP2P [33] and vid2vid-zero [66] manipulate the cross-attention map, while Dreamix [37] proposes an image-video mixed fine-tuning strategy. However, their generation or editing processes solely rely on text-based instructions [44, 45]. A subsequent work, Gen-1 [16], integrates depth maps alongside texts using cross-attention mechanisms to provide structural guidance. Both MCDiff [8] and LaMD [27] target motion-guided video generation; the former focuses on generating human action videos and encodes the dynamics by tracking the keypoints and reference points, while the latter employs a learnable motion latent to improve quality. Nevertheless, incorporating the guidance from efficient motion vectors or incorporating multiple guiding conditions within a single model is seldom explored in the general video synthesis field.

**Motion modeling.** Motion cues play a crucial role in video understanding fields, such as action recognition [1, 4, 6, 43, 60, 63, 64], action detection [10, 68, 77, 82], human video generation [39, 41, 67], *etc*. Pioneering works [1, 6, 39, 43, 63, 67] usually leverage hand-crafted dense optical flow [76] to embed motion information or design various temporal structures to encode long-range temporal representations. Due to the high computational demands of optical flow extraction, several attempts in compressed video recognition [7, 52, 69, 78] have begun to utilize more efficient motion vectors as an alternative to represent motions and have shown promising performance. In contrast to these works, we delve into the role of motions in video synthesis and demonstrate that motion vectors can enhance temporal controllability through a well-designed architecture.

## 3 VideoComposer

In this section, we will comprehensively present `VideoComposer` to showcase how it can enhance the controllability of video synthesis and enable the creation of highly customized videos. Firstly, we in brief introduce Video Latent Diffusion Models (VLDMs) upon which VideoComposer is designed, given their impressive success in various generative tasks. Subsequently, we delve into the details of

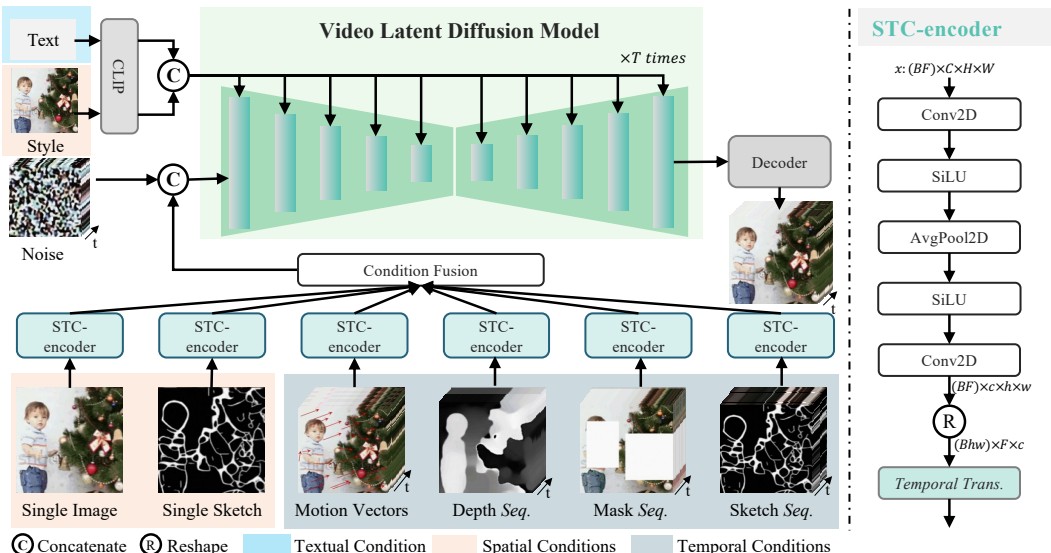

Figure 2: **Overall architecture** of `VideoComposer`. First, a video is decomposed into three types of conditions, including textual condition, spatial conditions and temporal conditions. Then, we feed these conditions into the unified STC-encoder or the CLIP model to embed control signals. Finally, the resulting conditions are leveraged to jointly guide VLDMs for denoising.

`VideoComposer`'s architecture, including the composable conditions and unified Spatio-Temporal Condition encoder (STC-encoder) as illustrated in Fig. 2. Finally, the concrete implementations, including the training and inference processes, will be analyzed.

### 3.1 Preliminaries

Compared to images, processing video requires substantial computational resources. Intuitively, adapting image diffusion models that process in the pixel space [40, 46] to the video domain impedes the scaling of `VideoComposer` to web-scale data. Consequently, we adopt a variant of LDMs that operate in the latent space, where local fidelity could be maintained to preserve the visual manifold.

**Perceptual video compression.** To efficiently process video data, we follow LDMs by introducing a pre-trained encoder [15] to project a given video $\boldsymbol{x} \in \mathbb{R}^{F \times H \times W \times 3}$ into a latent representation $\boldsymbol{z} = \mathcal{E}(\boldsymbol{x})$, where $\boldsymbol{z} \in \mathbb{R}^{F \times h \times w \times c}$. Subsequently, a decoder $\mathcal{D}$ is adopted to map the latent representations back to the pixel space $\bar{\boldsymbol{x}} = \mathcal{D}(\boldsymbol{z})$. We set $H/h = W/w = 8$ for rapid processing.

**Diffusion models in the latent space.** To learn the actual video distribution $\mathbb{P}(x)$, diffusion models [23, 54] learn to denoise a normally-distributed noise, aiming to recover realistic visual content. This process simulates the reverse process of a Markov Chain of length $T$. $T$ is set to 1000 by default. To perform the reverse process on the latent, it injects noise to $\boldsymbol{z}$ to obtain a noise-corrupted latent $\boldsymbol{z}_t$ following [48]. Subsequently, we apply a denoising function $\epsilon_\theta(\cdot, \cdot, t)$ on $\boldsymbol{z}_t$ and selected conditions $\boldsymbol{c}$, where $t \in \{1, ..., T\}$. The optimized objective can be formulated as:

$$\mathcal{L}_{VLDM} = \mathbb{E}_{\mathcal{E}(x), \epsilon \in \mathcal{N}(0,1), \boldsymbol{c}, t} \left[ \|\epsilon - \epsilon_\theta(\boldsymbol{z}_t, \boldsymbol{c}, t)\|_2^2 \right] \tag{1}$$

To exploit the inductive bias of locality and temporal inductive bias of sequentiality during denoising, we instantiate $\epsilon_\theta(\cdot, \cdot, t)$ as a 3D UNet augmented with temporal convolution and cross-attention mechanism following [25, 49, 62].

### 3.2 VideoComposer

**Videos as composable conditions.** We decompose videos into three distinct types of conditions, *i.e.*, textual conditions, spatial conditions and crucially temporal conditions, which can jointly determine the spatial and temporal patterns in videos. Notably, `VideoComposer` is a generic compositional framework. Therefore, more customized conditions can be incorporated into `VideoComposer` depending on the downstream application and are not limited to the decompositions listed above.

*Textual condition.* Textual descriptions provide an intuitive indication of videos in terms of coarse-grained visual content and motions. In our implementation, we employ the widely used pre-trained text encoder from OpenCLIP ViT-H/14 to obtain semantic embeddings of text descriptions.

*Spatial conditions.* To achieve fine-grained spatial control and diverse stylization, we apply three spatial conditions to provide structural and stylistic guidance: *i)* Single image. Video is made up of consecutive images, and a single image usually reveals the content and structure of this video. We select the first frame of a given video as a spatial condition to perform image-to-video generation. *ii)* Single sketch. We extract sketch of the first video frame using PiDiNet [55] as the second spatial condition and encourage `VideoComposer` to synthesize temporal-consistent video according to the structure and texture within the single sketch. *iii)* Style. To further transfer the style from one image to the synthesized video, we choose the image embedding as the stylistic guidance, following [3, 28]. We apply a pre-trained image encoder from OpenCLIP ViT-H/14 to extract the stylistic representation.

*Temporal conditions.* To accomplish finer control along the temporal dimension, we introduce four temporal conditions: *i)* Motion vector. Motion vector as a video-specific element is represented as two-dimension vectors, *i.e.*, horizontal and vertical orientations. It explicitly encodes the pixel-wise movements between two adjacent frames, as visualized by red arrows in Fig. 3. Due to the natural properties of motion vector, we treat this condition as a motion control signal for temporal-smooth synthesis. Following [52, 69], we extract motion vectors in standard

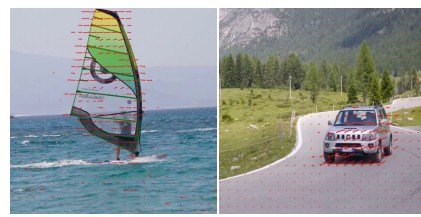

Figure 3: **Examples of motion vectors**.

MPEG-4 format from compressed videos. *ii)* Depth sequence. To introduce depth information, we utilize the pre-trained model from [47] to extract depth maps of video frames. *iii)* Mask sequence. To facilitate video regional editing and inpainting, we manually add masks. We introduce tube masks [17, 57] to mask out videos and enforce the model to predict the masked regions based on observable information. *iv)* Sketch sequence. Compared with the single sketch, sketch sequence can provide more control details and thus achieve precisely customized synthesis.

**STC-encoder.** Sequential conditions contain rich and complex space-time dependencies, posing challenges for controllable guidance. In order to enhance the temporal awareness of input conditions, we design a Spatio-Temporal Condition encoder (STC-encoder) to incorporate the space-time relations, as shown in Fig. 2. Specifically, a light-weight spatial architecture consisting of two 2D convolutions and an average pooling layer is first applied to the input sequences, aiming to extract local spatial information. Subsequently, the resulting condition sequence is fed into a temporal Transformer layer [61] for temporal modeling. In this way, STC-encoder facilitates the explicit embedding of temporal cues, allowing for a unified condition interface for diverse inputs, thereby enhancing inter-frame consistency. It is worth noting that we repeat the spatial conditions of a single image and single sketch along the temporal dimension to ensure their consistency with temporal conditions, hence facilitating the condition fusion process.

After processing the conditions by STC-encoder, the final condition sequences are all in an identical spatial shape to $z_t$ and then fused by element-wise addition. Finally, we concatenate the merged condition sequence with $z_t$ along the channel dimension as control signals. For textual and stylistic conditions organized as a sequence of embeddings, we utilize the cross-attention mechanism to inject textual and stylistic guidance.

### 3.3 Training and inference

**Two-stage training strategy.** Although `VideoComposer` can initialize with the pre-training of LDMs [48], which mitigates the training difficulty to some extent, the model still struggles in learning to simultaneously handle temporal dynamics and synthesize video content from multiple compositions. To address this issue, we leverage a two-stage training strategy to optimize `VideoComposer`. Specifically, the first stage targets pre-training the model to specialize in temporal modeling through text-to-video generation. In the second stage, we optimize `VideoComposer` to excel in video synthesis controlled by the diverse conditions through compositional training.

**Inference.** During inference, DDIM [80] is employed to enhance the sample quality and improve inference efficiency. We incorporate classifier-free guidance [22] to ensure that the generative results

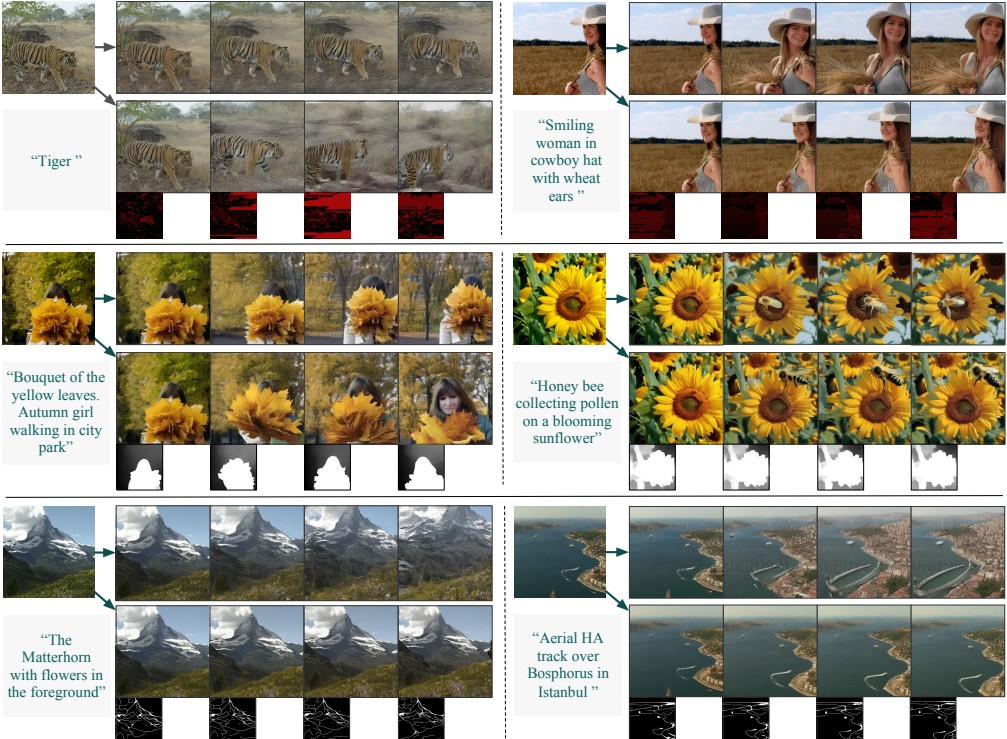

Figure 4: **Compositional image-to-video generation**. We showcase six examples, each displaying two generated videos. The upper video is generated using a given single frame as the spatial condition and a textual condition describing the scene. The lower video is generated by incorporating an additional sequence of temporal conditions to facilitate finer control over the temporally evolving structure.

adhere to specified conditions. The generative process can be formalized as:

$$\hat{\epsilon}_\theta(z_t, c, t) = \epsilon_\theta(z_t, c_1, t) + \omega \left( \epsilon_\theta(z_t, c_2, t) - \epsilon_\theta(z_t, c_1, t) \right) \qquad (2)$$

where $\omega$ is the guidance scale; $c_1$ and $c_2$ are two sets of conditions. This guidance mechanism extrapolates between two condition sets, placing emphasis on the elements in $(c_2 \setminus c_1)$ and empowering flexible application. For instance, in text-driven video inpainting, $c_2$ represents the expected caption and a masked video, while $c_1$ is an empty caption and the same masked video.

## 4 Experiments

### 4.1 Experimental setup

**Datasets.** To optimize `VideoComposer`, we leverage two widely recognized and publicly accessible datasets: WebVid10M [2] and LAION-400M [51]. WebVid10M [2] is a large-scale benchmark scrapped from the web that contains 10.3M video-caption pairs. LAION-400M [51] is an image-caption paired dataset, filtered using CLIP [44].

**Evaluation metrics.** We utilize two metrics to evaluate `VideoComposer`: *i)* To evaluate video continuity, we follow Gen-1 [16] to compute the average CLIP cosine similarity of two consecutive frames, serving as a **frame consistency metric**; *ii)* To evaluate motion controllability, we adopt end-point-error [56, 73] as a **motion control metric**, which measures the Euclidean distance between the predicted and the ground truth optical flow for each pixel.

### 4.2 Composable video generation with versatile conditions

In this section, we demonstrate the ability of `VideoComposer` to tackle various tasks in a controllable and versatile manner, leveraging its inherent compositionality. It's important to note that the conditions employed in these examples are customizable to specific requirements. We also provide additional results in the supplementary material for further reference.

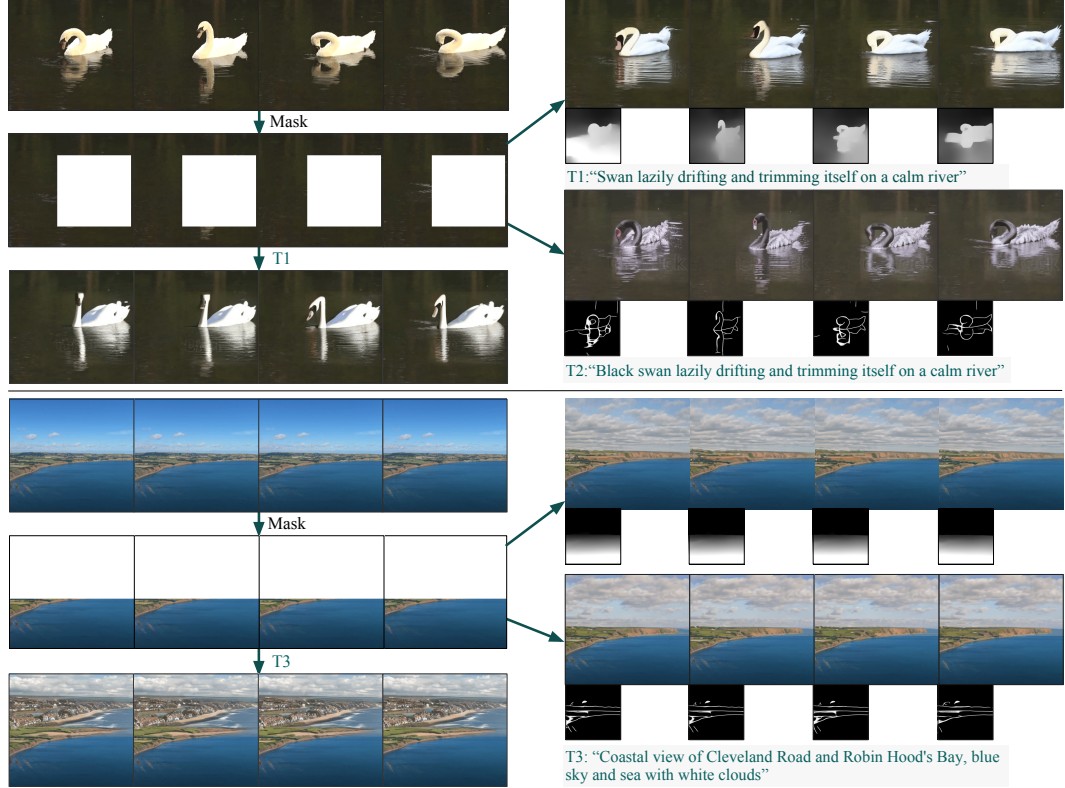

Figure 5: **Compositional video inpainting.** By manually adding masks to videos, `VideoComposer` can perform video inpainting, facilitating the restoration of the corrupted parts according to textual instructions. Furthermore, by incorporating temporal conditions specifying the visual structure, `VideoComposer` can perform customized inpainting that conforms to the prescribed structure.

**Compositional Image-to-video generation.** Compositional training with a single image endows `VideoComposer` with the ability of animating static images. In Fig. 4, we present six examples to demonstrate this ability. `VideoComposer` is capable of synthesizing videos conformed to texts and the initial frame. To further obtain enhanced control over the structure, we can incorporate additional temporal conditions. We observe resultant videos consistently adhere to the given conditions.

**Compositional video inpainting.** Jointly training with masked video endows the model with the ability of filling the masked regions with prescribed content, as shown in Fig. 5. `VideoComposer` can replenish the mask-corrupted regions based on textual descriptions. By further incorporating temporal conditions, *i.e*, depth maps and sketches, we obtain more advanced control over the structure.

**Compositional sketch-to-video generation.** Compositional training with single sketch empowers `VideoComposer` with the ability of animating static sketches, as illustrated in Fig. 6. We observe that `VideoComposer` synthesizes videos conforming to texts and the initial sketch. Furthermore, we observe that the inclusion of mask and style guidance can facilitate structure and style control.

## 4.3 Comparative experimental results

**Depth-to-video controllability comparison.** In Fig. 7, we compare our VideoComposer with Text2Video-Zero [30] and existing state-of-the-art Gen-1 [16]. We observed that Text2Video-Zero suffers from appearance inconsistency and structural flickering due to the lack of temporal awareness. Meanwhile, Gen-1 produces a video with color inconsistency and structure misalignment (revealed by the orientation of the bird head). The video generated by VideoComposer is faithful to the structure of the input depth sequence and maintains a continuous appearance. This shows the superiority of our VideoComposer in terms of controllability.

**Text-to-video generation performance.** Although `VideoComposer` is not specifically tailored for text-to-video generation, its versatility allows `VideoComposer` to perform the traditional text-to-video generation task effectively. In Tab. 1, we follow the evaluation settings in Video LDM [5] to adopt

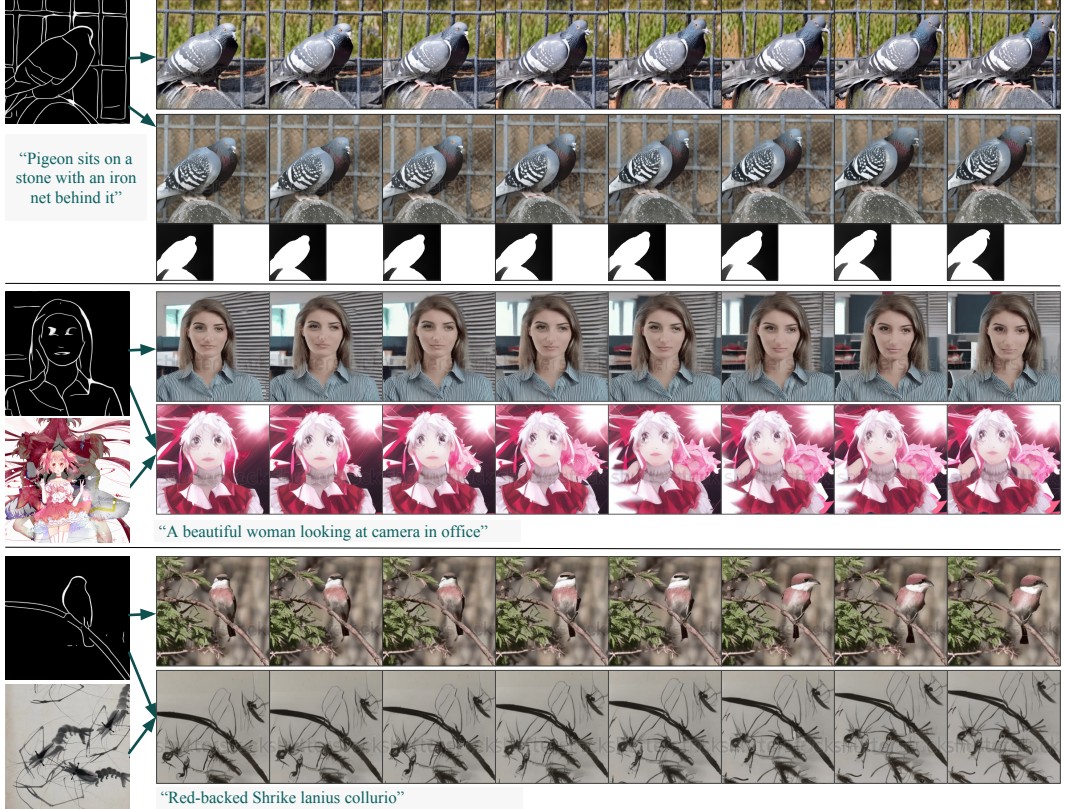

Figure 6: **Compositional sketch-to-video generation**. In the first example, the upper video is generated using text and a single sketch as the conditions, while the lower is generated by using an additional mask sequence for finer control over the temporal patterns. For the last two examples, the upper video is generated using a single sketch and a textual condition, while the lower is generated with an additional style from a specified image.

Table 1: **Text-to-video generation performance** on MSR-VTT.

| Method | Zero-shot | FVD ↓ | CLIPSIM ↑ |
|---|---|---|---|
| GODIVA [70] | No | - | 0.2402 |
| Nüwa [71] | No | - | 0.2439 |
| CogVideo (Chinese) [26] | Yes | - | 0.2614 |
| CogVideo (English) [26] | Yes | 1294 | 0.2631 |
| MagicVideo [83] | Yes | 1290 | - |
| Make-A-Video [53] | Yes | - | **0.3049** |
| Video LDM [5] | Yes | - | 0.2929 |
| Text-to-video pre-training (First stage) | Yes | 803 | 0.2876 |
| VideoComposer | Yes | **580** | 0.2932 |

Fréchet Video Distance (FVD) and CLIP Similarity (CLIPSIM) as evaluation metrics and present the quantitative results of text-to-video generation on MSR-VTT dataset compared to other existing methods. The results in the table demonstrate that `VideoComposer` achieves competitive performance compared to state-of-the-art text-to-video approaches. In addition, `VideoComposer` outperforms our first-stage text-to-video pre-training, demonstrating that `VideoComposer` can achieve compositional generation without sacrificing its capability of text-to-video generation. In the future, we aim to advance `VideoComposer` by leveraging stronger text-to-video models for more powerful synthesis.

## 4.4 Experimental results of motion control

**Quantitative evaluation.** To validate superior motion controllability, we utilize the motion control metric. We randomly select 1000 caption-video pairs and synthesize corresponding videos. The results are presented in Tab. 2. We observe that the inclusion of motion vectors as a condition reduce the motion control error, indicating an enhancement of motion controllability. The incorporation of STC-encoder further advances the motion controllability.

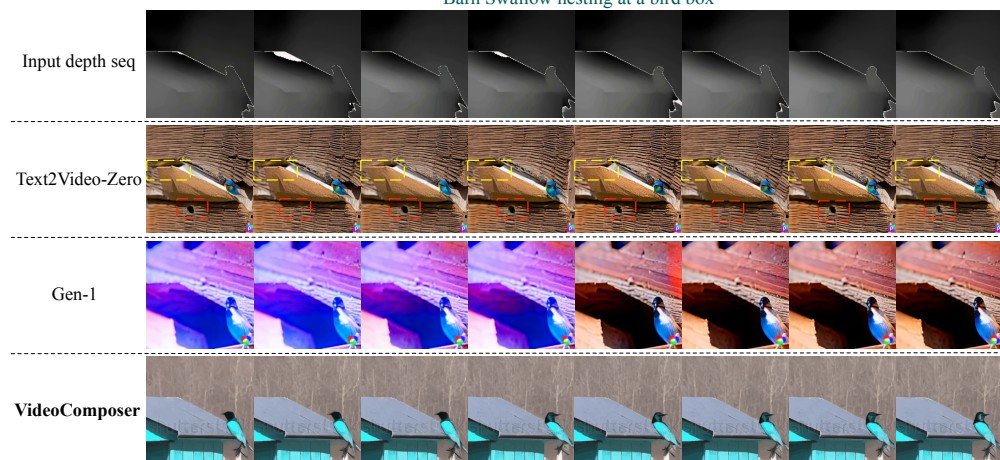

"Barn Swallow nesting at a bird box"

Input depth seq

Text2Video-Zero

Gen-1

**VideoComposer**

Figure 7: Comparison with methods for controllable video generation. Results are generated utilizing the textual condition and the depth sequence in the first row. Text2Video-Zero suffers from appearance inconsistency and structural flickering due to the lack of temporal awareness. Gen-1 produces a video with color inconsistency and structural misalignment. The video generated by VideoComposer is faithful to the structure of the input depth sequence and maintains a continuous appearance.

"American Staffordshire Terrier is lying on the floor in an abandoned building"

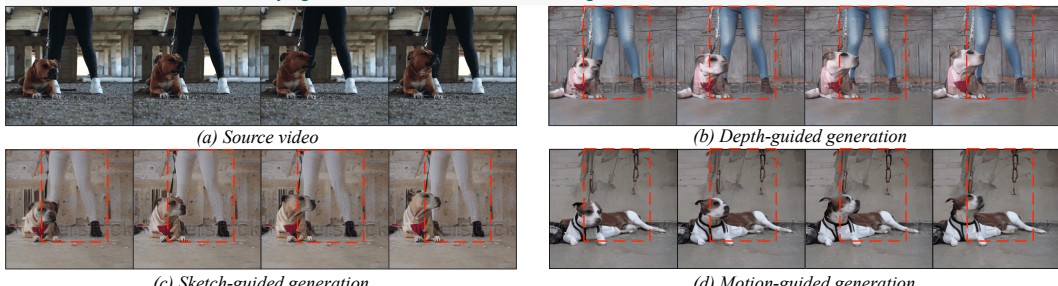

*(a) Source video*    *(b) Depth-guided generation*

*(c) Sketch-guided generation*    *(d) Motion-guided generation*

Figure 8: **Video-to-video translation**. We extract a sequence of depth maps, sketches or motion vectors from the source video, along with textual descriptions, to perform the translation. By utilizing motion vectors, we achieve **static-background removal**.

**Motion vectors prioritizing moving visual cues.** Thanks to the nature of motion vectors, which encode inter-frame variation, static regions within an image are inherently omitted. This prioritization of moving regions facilitates motion control during synthesis. In Fig. 8, we present results of video-to-video translation to substantiate such superiority. We observe that motion vectors exclude the static

Table 2: **Evaluating the motion controllability**. "Text" and "MV" represent the utilization of text and motion vectors as conditions for generation.

| Method | Text | MV | Motion control ↓ |
|---|---|---|---|
| *w/o* STC-encoder | ✓ | | 4.03 |
| *w/o* STC-encoder | ✓ | ✓ | 2.67 |
| VideoComposer | ✓ | ✓ | **2.18** |

background, *i.e.*, human legs, a feat that other temporal conditions such as depth maps and sketches cannot accomplish. This advantage lays the foundation for a broader range of applications.

**Versatile motion control with motion vectors.** Motion vectors, easily derived from hand-crafted strokes, enable more versatile motion control. In Fig. 9, we present visualization comparing CogVideo [26] and VideoComposer. While CogVideo is limited to insufficient text-guided motion control, VideoComposer expands this functionality by additionally leveraging motion vectors derived from hand-crafted strokes to facilitate more flexible and precise motion control.

## 4.5 Ablation study

In this subsection, we conduct qualitative and quantitative analysis on VideoComposer, aiming to demonstrate the effectiveness of incorporating STC-encoder.

**Quantitative analysis.** In Tab. 3, we present the frame consistency metric computed on 1000 test videos. We observe that incorporating STC-encoder augments the frame consistency, which we

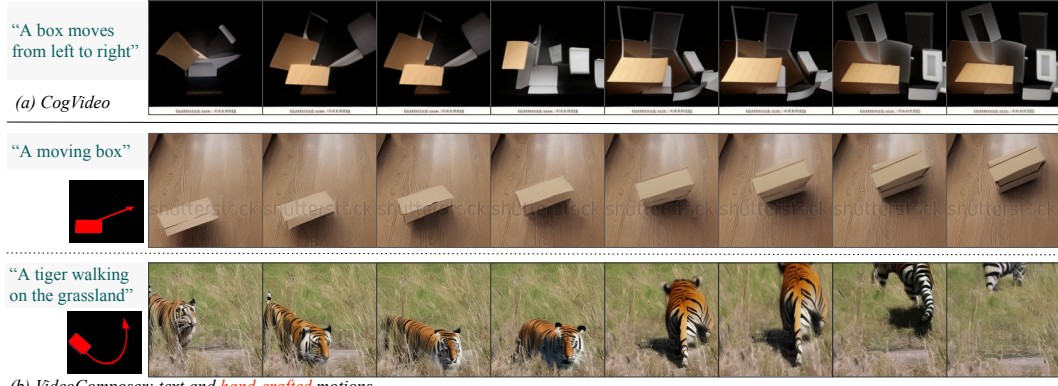

(a) CogVideo

"A box moves from left to right"

"A moving box"

"A tiger walking on the grassland"

(b) *VideoComposer: text and hand-crafted motions*

Figure 9: **Versatile motion control using hand-crafted motions.** **(a)** Limited motion control using CogVideo [26]. **(b)** Fine-grained and flexible motion control, empowered by VideoComposer.

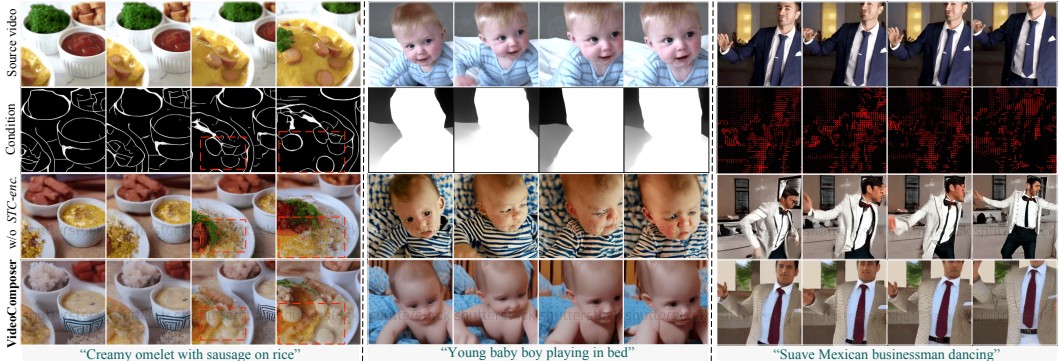

"Creamy omelet with sausage on rice"  "Young baby boy playing in bed"  "Suave Mexican businessman dancing"

Figure 10: **Qualitative ablation study**. We present three representative examples. The last two rows of videos display generated videos conditioned on a textual condition and one additional temporal condition (*i.e.*, sketches, depth maps or motion vectors). Regions exhibiting deficiencies or fidelity are emphasized within red boxes.

attribute to its temporal modeling capacity. This observation holds for various temporal conditions such as sketches, depth maps and motion vectors.

**Qualitative analysis.** In Fig. 10, we exemplify the usefulness of STC-encoder. We observe that in the first example, videos generated by VideoComposer without STC-encoder generally adhere to the sketches but omit certain detailed information, such as several round-shaped ingredients. For the left two examples, VideoComposer without STC-encoder generates videos that are structurally inconsistent with conditions. We can also spot the noticeable defects in terms of human faces and poses. Thus, all the above examples can validate the effectiveness of STC-encoder.

Table 3: **Quantitative ablation study of STC-encoder**. "Conditions" denotes the conditions utilized for generation.

| Method | Conditions | Frame consistency ↑ |
|---|---|---|
| *w/o* STC-encoder | Text and sketch sequence | 0.910 |
| VideoComposer | | **0.923** |
| *w/o* STC-encoder | Text and depth sequence | 0.922 |
| VideoComposer | | **0.928** |
| *w/o* STC-encoder | Text and motion vectors | 0.915 |
| VideoComposer | | **0.927** |

## 5 Conclusion

In this paper, we present VideoComposer, which aims to explore the compositionality within the realm of video synthesis, striving to obtain a flexible and controllable synthesis system. In particular, we explore the use of temporal conditions for videos, specifically motion vectors, as powerful control signals to provide guidance in terms of temporal dynamics. An STC-encoder is further designed as a unified interface to aggregate the spatial and temporal dependencies of the sequential inputs for inter-frame consistency. Our experiments, which involve the combination of various conditions to augment controllability, underscore the pivotal role of our design choices and reveal the impressive creativity of the proposed VideoComposer.

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

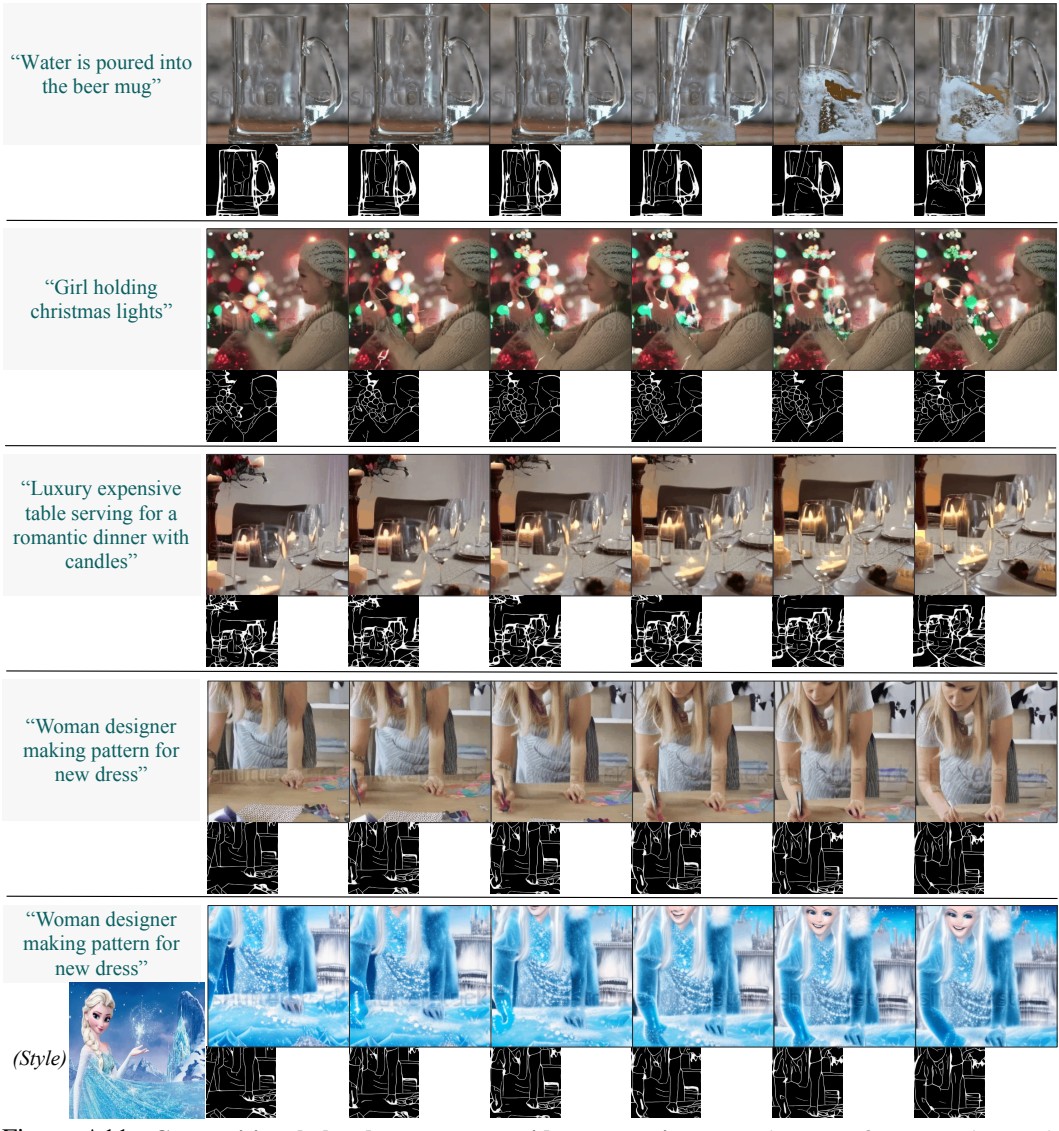

Figure A11: **Compositional sketch sequence-to-video generation**. We showcase five examples, each displaying a video generated from a sequence of sketches and a textual description. The final example additionally incorporates a style condition.

# Appendix

In this Appendix, we first elaborate on more implementation details (Appendix A) and present more experimental results (Appendix B). Next, we provide a section of discussion (Appendix C) on the limitations and potential societal impact of `VideoComposer`.

## A  More implementation details

**Pre-training details.** We adopt AdamW [35] as the default optimizer with a learning rate set to $5 \times 10^{-5}$. In total, `VideoComposer` is pre-trained for 400k steps, with the first and second stage being pre-trained for 132k steps and 268k steps, respectively. In terms of two-stage pre-training, we allocate one fourth of GPUs to perform image pre-training, while the rest of the GPUs are dedicated to video pre-training. We use center crop and randomly sample video frames to compose the video input whose $F = 16$, $H = 256$ and $W = 256$. During the second stage pre-training, we adhere to [28], using a probability of $0.1$ to keep all conditions, a probability of $0.1$ to discard all conditions, and an independent probability of $0.5$ to keep or discard a specific condition. Regarding the use of

WebVid10M [2], we sample frames from videos using various strides to ensure frame rate equal to 4, aiming to maintain a consistent frame rate. We train `VideoComposer` jointly on video-text and image-text pairs by treating images as 'one-frame' videos. When using image-text pairs for training, the shape of the input noise is $1 \times h \times w \times c$, where the temporal length is 1 (*i.e.*, $F = 1$). In experiments, we use separate STC-encoders for different conditions without weight sharing. In our ablation study of STC-encoder, the baseline method (*w/o* STC-encoder) entails removing the temporal Transformer in STC-encoder while retaining the spatial convolution, designed to verify the effectiveness of incorporating temporal modeling. The spatial convolution remains in place to ensure the dimensions of all input conditions are consistent.

**The structure of 3D UNet as** $\epsilon_\theta(\cdot, \cdot, t)$**.** To leverage the benefits of LDMs pre-trained on web-scale image data, *i.e.*, Stable Diffusion[2], we extend the 2D UNet to a 3D UNet by introducing temporal modeling layers. Specifically, within a single UNet block, we employ four essential building blocks: spatial convolution, temporal convolution, spatial transformer and temporal transformer. The spatial blocks are inherited from LDMs, while temporal processing blocks are newly introduced. Regarding temporal convolution, we stack four convolutions with $1 \times 1 \times 3$ kernel, ensuring the temporal receptive field is ample for capturing temporal dependencies; regarding temporal transformer, we stack one Transformer layer and accelerate its inference using flash attention [12].

**More details about two-stage training strategy.** The two-stage training strategy utilized in VideoComposer is designed to methodologically address the learning challenge. In the first stage, VideoComposer focuses on learning the temporal dynamics by only leveraging the textual condition. This foundation allows for a focused understanding of temporal relationships within the video content. In the second stage, VideoComposer builds on the temporal modeling ability acquired from the first stage to perform compositional training. In this stage, it extends its learning by utilizing all three kinds of conditions: textual, spatial, and temporal conditions. The major difference between the two stages lies in this incorporation of additional conditions, leading to a comprehensive learning of synthesizing video content from multiple compositions.

# B  More experimental results

In this subsection, we aim to provide additional experiments that complement the findings presented in the main paper and showcase more versatile controlling cases.

**Compositional sketch sequence-to-video generation.** Compositional training with sketch sequences enables `VideoComposer` to possess the ability of generation videos adhering to sketch sequences. This generation paradigm lays more emphasis on the structure control, which differs from compositional sketch-to-video generation and can be viewed as video-to-video translation. In Fig. A11, we exemplify this capacity. We observe videos' fidelity to the provided conditions, including texts, sketches and style.

**Compositional depth sequence-to-video generation.** Conducting compositional training with depth sequences allows `VideoComposer` to effectively generate videos in accordance with depth sequences. In Fig. A12, we illustrate this capability. Videos generated with `VideoComposer` faithfully adhere to the given conditions, including text prompts, depth maps, and style.

**Motion transfer.** Incorporating motion vectors as a composition of videos enables motion transferability. In Fig. A13, we conduct experiments to demonstrate such capability. Through utilizing hand-crafted motion vectors or motion vectors extracted from off-the-shelf source videos, we can transfer the motion patterns to synthesized videos.

# C  Discussion

**Limitations.** Due to the absence of a publicly available large-scale and high-quality dataset, we have developed `VideoComposer` using the watermarked WebVid10M dataset. As a result, the synthesized videos contain watermarks, which affect the generation quality and lead to less visually appealing results. Furthermore, in order to reduce the training cost, the resolution of the generated videos is limited to 256×256. Consequently, some delicate details might not be sufficiently clear. In the future,

---

[2]https://github.com/Stability-AI/stablediffusion

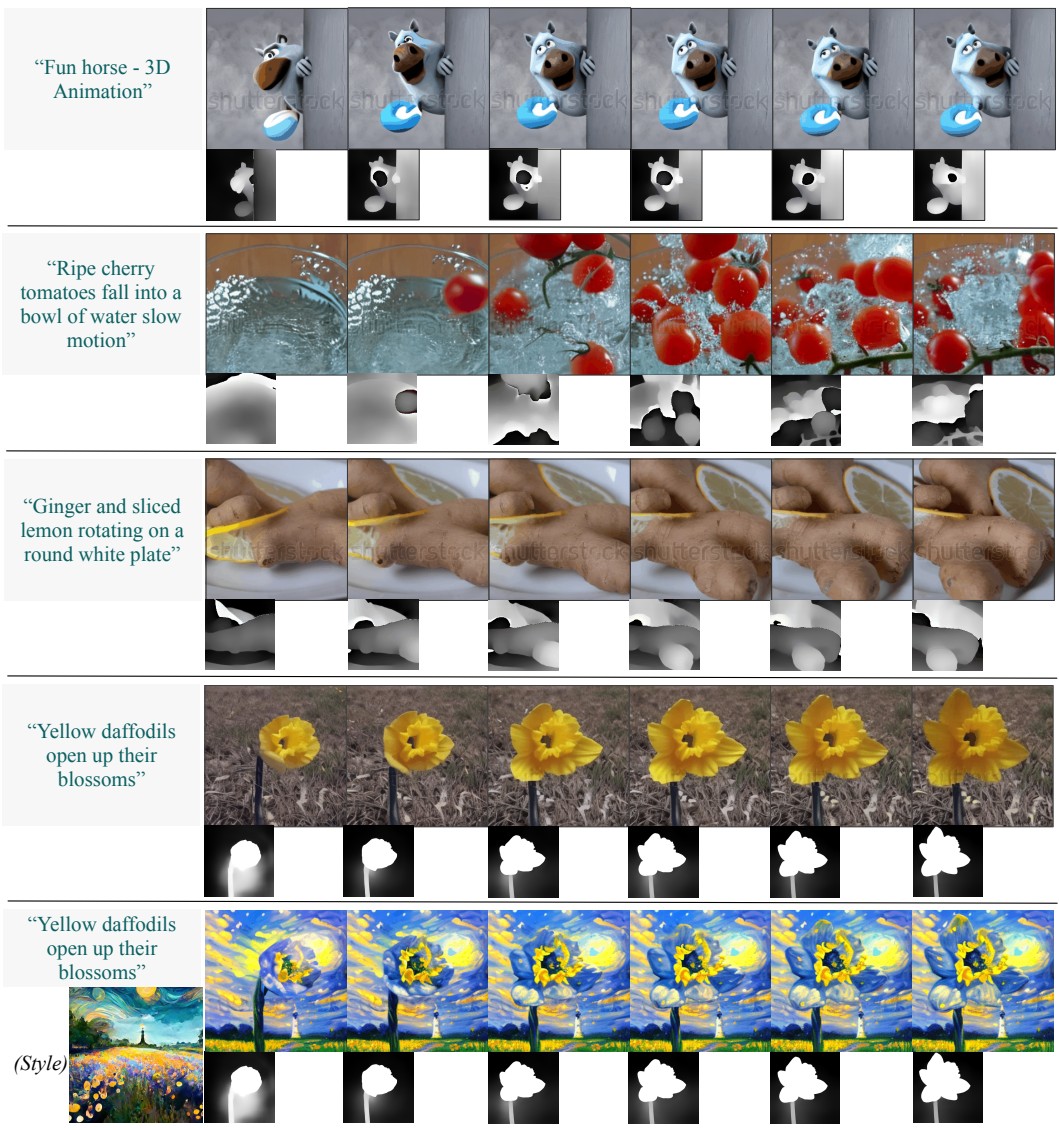

Figure A12: **Compositional depth sequence-to-video generation**. We showcase five examples, each displaying a video generated from a sequence of depth maps and a textual description. The final example additionally incorporates a style condition.

we plan to utilize super-resolution models to expand the resolution of the generated videos to improve the visual quality.

**Potential societal impact.** `VideoComposer`, as a generic video synthesis technology, possesses the potential to revolutionize the content creation industry, offering unprecedented flexibility and creativity, and hence, promising significant commercial advantages. Traditional content creation processes are labor- and cost-intensive. `VideoComposer` could alleviate these burdens by enabling designers to manipulate subjects, styles, and scenes through instructions spanning human-written text, and styles and subjects sourced from other images. Moreover, `VideoComposer` could potentially revolutionize education industry by creating unique and customized video scenarios for teaching complex concepts.

However, it's necessary to note that `VideoComposer` also represents a dual-use technology with inherent risks to society. As with prior generative foundation models, such as Imagen Video [24] and Make-A-Video [53], `VideoComposer` inherits the implicit knowledge embedded within the pre-trained model (*i.e.*, StableDiffusion) and the pre-trained dataset (*i.e.*, WebVid and LAION). Potential issues include but not limited to the propagation of social biases (such as gender and racial bias) and the creation of offensive content.

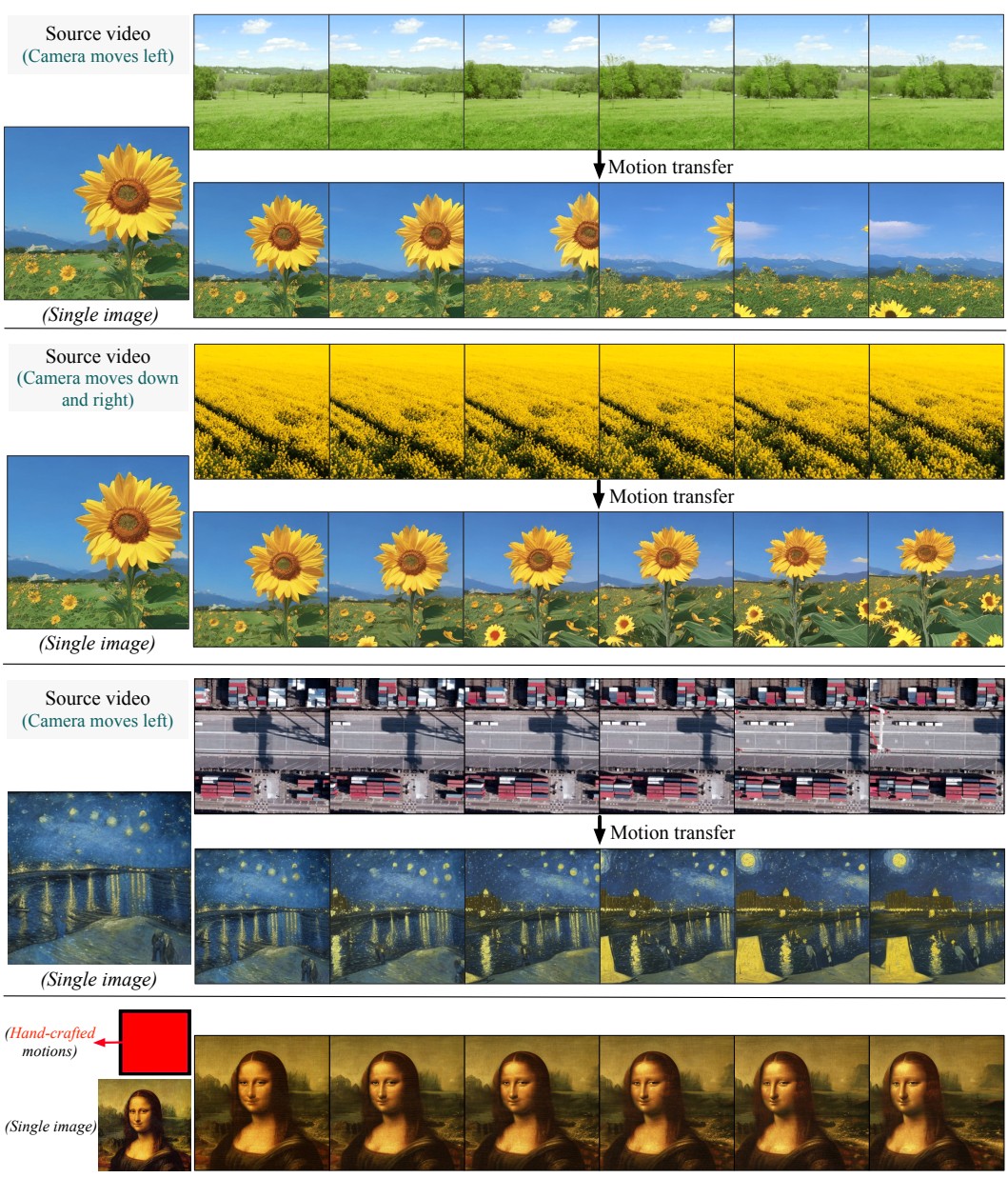

Figure A13: **Motion transfer.** We showcase four examples, each displaying a video generated from a single image and motions. In the first three examples, we transfer the motion patterns in a source video to the generated video by extracting and utilizing motion vectors. The final example incorporates hand-crafted motions instead.

Given that `VideoComposer` is a research-oriented project aimed at investigating compositionality in diffusion-based video synthesis, our primary focus lies in scientific exploration and proof of concept. If `VideoComposer` is deployed beyond the scope of research, we strongly recommend several precautionary measures to ensure its responsible and ethical use: **(i)** Rigorous evaluation and oversight of the deployment context should be conducted; **(ii)** Necessary filtering of prompts and generated content should be implemented to prevent misuse.

