# OpenReview forum: "VideoComposer: Compositional Video Synthesis with Motion Controllability"
_NeurIPS.cc/2023/Conference — NeurIPS 2023 poster_

### Official Review · Reviewer_Pp5z · 2023-07-02

**Soundness:** 3 good
**Presentation:** 2 fair
**Contribution:** 3 good
**Rating:** 6
**Confidence:** 4

**Summary:**

This work aims to allows users to flexibly compose a video with textual conditions, spatial conditions, and temporal conditions.
It introduces a novel framework namely VideoComposer based on the paradigm of compositional generation.
To be specific, it introduces the motion vector from compressed videos as an explicit control signal to provide guidance regarding temporal dynamics.
Moreover, it develop a Spatio-Temporal Condition encoder (STC-encoder) that serves as a unified interface to effectively incorporate the spatial and temporal relations of sequential inputs, with which the model could make better use of temporal conditions and hence achieve higher inter-frame consistency.
Extensive experiments demonstrate that VideoComposer control the spatial and temporal patterns simultaneously within a synthesized video in various forms.

**Strengths:**

1. It introduces motion vector as a more flexible user-guided signal.
2. It proposes Spatio-Temporal Condition encoder (STC-encoder) that serves as a unified interface to effectively incorporate the spatial and temporal relations of sequential inputs.
3. Extensive experiments show the effectiveness and superiority of VideoComposer.

**Weaknesses:**

1. What is the difference between the roles of the ``Style`` of CLIP and ``Single Image`` of STC-encoder? They both seem to provide content to videos.
2. VideoComposer only obtain comparable performance with prior video generative models. Is it more efficient than previous methods? The authors could give their comparisons in training cost and inference time.
3. Lack of extensive visualization comparisons with existing video generative models. The authors are encouraged to provide extensive qualitative comparisons in video generation task.


**Questions:**

See weakness.

**Limitations:**

Yes.

---

> ### Author Rebuttal · Authors · 2023-08-09
>
>
> **Q1: What is the difference between the roles of the Style of CLIP and Single Image of STC-encoder? They both seem to provide content to videos.**
>
>  Thank you for highlighting this point.
> - **Style condition**.  The style condition mainly encapsulates the holistic characteristics of the input image, capturing elements like its color, style, and content.  Analogous to the textual condition, it functions as a **global** descriptor, providing a generalized perspective of the image. When utilizing an image as this global condition, spatial pooling is employed to derive a singular embedding.  This spatial reduction, however, causes the loss of detailed structural information, which can lead to generalized content interpretation. Thus, it is possible that the generated video can have varying poses (if containing humans or animals) and slight color shifts compared to the reference image.
> - **Single image condition**. The primary role of the single image condition is to serve as the initial frame for the video being generated.  As such, it conveys **local** details and intricacies of the image, ensuring the video's initiation is aligned pixel-for-pixel with the provided image.
>
> In conclusion, both conditions influence the video's content and color, but the depth and granularity of influence differ.
>
>
>
> **Q2: VideoComposer only obtain comparable performance with prior video generative models. Is it more efficient than previous methods? The authors could give their comparisons in training cost and inference time.**
>
> Thank you for raising this point. We want to clarify that the primary objective of VideoComposer is to enhance controllability and applicability, rather than to reduce training cost and inference time. Recognizing the value of the reviewer's concern, we also provide a brief discussion in terms of **efficiency**.
> - **Controllability and versatility**. One of the primary advantages of VideoComposer over previous methods is its augmented controllability. Both Make-A-Video and Video LDM primarily support video generation from text descriptions, limiting their capacity for customized, controllable video generation. In contrast, VideoComposer benefits from diverse conditional guidances, facilitating video generation utilizing user-defined textual, spatial, and temporal conditions. This versatility paves a path to truly compositional video generation tailored to specific requirements.
> - **Broad applicability**. VideoComposer's design isn't just about controllability, but also about applicability.  Our model's capability to address multiple video generation tasks without the need for repeated re-training underscores its unique value. We have equipped one model with a multitude of application scenarios, marking a significant superiority to existing works.
> - **Efficiency**. While we can't provide direct numerical comparisons due to the unavailability of source codes from competitive methods like Make-A-Video and Video LDM, we'd like to highlight certain design features that contribute to VideoComposer's efficiency. Specifically, unlike Make-A-Video, which leverages multiple cascaded models, VideoComposer adopts a unified approach, possibly reducing both the training cost and inference time. We anticipate that this design could allow VideoComposer to achieve comparable efficiency to existing methods such as Video LDM since both base on Stable Diffusion and extend it by adding additional temporal layers.
>
> In summary, while we acknowledge the request for direct comparisons in efficiency, we'd like to emphasize VideoComposer's controllability and versatility.  These features make our model a crucial contribution to the current landscape of video synthesis.
>
> **Q3: Lack of extensive visualization comparisons with existing video generative models. The authors are encouraged to provide extensive qualitative comparisons in video generation task.**
>
> Thanks for the suggestion. We show more qualitative comparisons with existing Text2Video-Zero and Gen-1 in Figure R4. We observed that Text2Video-Zero suffers from appearance inconsistency and structural flickering due to the lack of temporal awareness. Meanwhile, Gen-1 produces a video with color inconsistency and structure misalignment (revealed by the orientation of the bird head). The video generated by VideoComposer is faithful to the structure of the input depth sequence and maintains a continuous appearance. The above experiments demonstrate the superiority of our method in terms of controllability.

---

> > ### Comment · Reviewer_Pp5z · 2023-08-10
> > **Thanks for reponse**
> >
> > Thanks for the author's elaborate response, and all my concerns have been well addressed.

---

> > > ### Author Response · Authors · 2023-08-17
> > > **Thanks for the reply**
> > >
> > > Dear Reviewer Pp5z,
> > >
> > > We really appreciate your constructive feedback to improve our manuscript, thank you!
> > >
> > > Best regards,
> > >
> > > The Authors

---

### Official Review · Reviewer_EXcw · 2023-07-02

**Soundness:** 3 good
**Presentation:** 3 good
**Contribution:** 3 good
**Rating:** 6
**Confidence:** 4

**Summary:**

This work proposes a new method called VideoComposer for conditional video generation, especially for video-to-video translation. VideoComposer is constructed upon the Video Latent Diffusion Model and introduces an STC-encoder to integrate multiple spatial and temporal conditions such as RGB images, sketches, motion vector sequences, etc. The architecture design involves simple 2D convolutions and temporal transformer layers. The conditional features are fed into the U-Net input together with noise. The demonstrated results have good temporal consistency.

**Strengths:**

- This is one of the pioneering works in controllable video synthesis. The temporal consistency of the video results is impressive, considering that its conditioning modeling enables several editing abilities such as image-to-video translation and motion/depth/sketch-driven local/global video editing.

- The jointly training strategy is good for flexible inference within one model, e.g., video inpainting, without second training.

- The paper organization and illustrations are easy to follow.

**Weaknesses:**

- The authors could have tried other design choices for integrating Condition Fusion as input into the U-Net, such as integration through cross-attention.

- In line 215, it is claimed that “we observe that the inclusion of mask and style guidance can facilitate structure and style control.” However, the corresponding evidence should be presented for the style representation extracted by clip image encoder and concatenated with text embedding.

- It seems that a single STC-encoder is used for all different conditions via random dropout. It would be interesting to see if different STC-encoder weights for different conditions are better.

- The examples in Figure 6 with reference image look like failure cases. Besides, the tiger texture and box shape are changed in Figure 8. It would be helpful to see more discussion and analysis on this part.

- The ablation study of STC-encoder is not presented in a fair way. The main benefit of using STC-encoder comes from the video information condition instead of the network design.

- The important comparisons and discussions with other methods are not sufficient, such as VideoP2P and vid2vid-zero mentioned in the related works.

**Questions:**

The comparisons with other methods are not sufficient and the ablation study is not well presented.

**Limitations:**

The societal impact has been discussed.

---

> ### Author Rebuttal · Authors · 2023-08-09
>
>
> **Q1: Integrating the Condition Fusion into U-Net through cross-attention.**
>
> The suggestion to use cross-attention for Condition Fusion in the U-Net is appreciated. While our current choice might not potentially be the optimal one, improving the micro-design of conditioning is beyond the scope of this work and requires considerable computational overhead associated with pre-training on LAION and WebVid. Therefore, we finalize this design choice more from the perspective of empirical analysis.
>
> In VideoComposer, regarding the injection of global conditions, such as textual condition and style condition, we opt for cross-attention mechanisms following Stable Diffusion as it contains the high-level and abstractive information. Due to the quadratic complexity of cross-attention mechanisms, implementing cross-attention for Condition Fusion (which is not pooled in the spatial dimension) introduces considerable computational overhead, especially when compared against the relatively efficient concatenation approach.
>
> **Q2: Evidence should be provided to prove the statement in Line 215 that "mask and style guidance can facilitate structure and style control".**
>
> We appreciate the reviewer's observation. Our description in line 215 might have been ambiguous. To align with this statement, we add the corresponding result in Figure R3 by adding the style condition. The generated video adheres to the given text, style and mask sequence. We will revise the confused descriptions in the next version. Feel free to raise questions if we misunderstand the question.
>
> **Q3: Is the STC-encoder shared across all conditions? Would the STC-encoder perform better with distinct weights?**
>
>  Thanks. We use separate STC-encoders for different conditions without weight sharing by default. In Figure R5, we compare the results using STC-encoders *w/* and *w/o* weight sharing on video inpainting, and find that the former causes performance degradation for conditions like mask sequence. We attribute this to the uniqueness of different conditions, and sharing weights may lead to modeling difficulties. We will further improve the description of the STC-encoder in the revision.
>
> **Q4: More analysis should be provided to clarify: (i) the examples in Figure 6 with reference image look like failure cases; (2) the tiger texture and box shape are changed in Figure 8.**
>
>  We appreciate your observation about Figure 6 and Figure 8.
>
> Firstly, We want to clarify that the role of the reference image is primarily as a global condition, offering stylistic features such as color and some aspects of content. As such, the generated videos tend to resemble the reference in terms of color and certain content attributes. According to our observation, we think the examples in Figure 6 exhibit such properties, thereby serving as successful cases.
>
> We acknowledge that in Figure 8, the tiger texture and box shapes are compromised as the video progresses. This stems from a lack of structural control since we only provide the motion condition and the textual condition. To enable temporally consistent generation, we augment the motion-controlled video generation with a simple structure control by adding an auxiliary single-sketch condition, as shown in Figure R6. This addition has substantially improved the shape consistency in the generated video, rendering results that outperform those presented in the main paper.
>
>
> **Q5: The ablation study of STC-encoder is not presented in a fair way. The main benefit of using STC-encoder comes from the video information condition instead of the network design.**
>
> Thank you for bringing our concern.
>
> We might fail to precisely communicate the setting of this ablation study. To clarify, in our ablation study of STC-encoder, the baseline method (*w/o* STC-encoder) entails removing the temporal Transformer in STC-encoder while retaining the spatial convolution, designed to verify the effectiveness of incorporating temporal modeling. The spatial convolution remains in place to ensure the dimensions of all input conditions are consistent.
>
> Under such a setting, VideoComposer and the baseline method both take advantage of the informative video conditions as input and, therefore, are compared in a fair way. As indicated in Line 242, such an ablation study underscores the significance of the STC-encoder's temporal modeling capacity during the input phase.
>
> In light of your feedback, we will supplement the details of the baseline method in the revised version to avoid confusion.
>
> **Q6: Comparisons and discussions with methods, such as VideoP2P and vid2vid-zero, are not sufficient.**
>
>  Thank you for raising this point. VideoP2P and vid2vid-zero are customized for video editing, which require the access to the original reference video (*i.e.*, the video to be edited) in order to finely optimize the video editing model for every video to be manipulated. However, our VideoComposer can circumvent this time-consuming re-training, and video generation can be performed given video conditions without accessing the original video.
>
> For a more illustrative comparison with VideoP2P and vid2vid-zero, we present results in Figure R7. In this example, our objective is to transform the dog in the video to a tiger using an updated textual condition. Even though VideoP2P and vid2vid-zero need to specially optimize the model with the reference video, they still have difficulty maintaining structural consistency due to the lack of sequential structure guidance. In contrast, the video edited by VideoComposer can retain the structural alignment with the reference video while ensuring temporal continuity.

---

> > ### Comment · Reviewer_EXcw · 2023-08-20
> >
> > Thanks for the rebuttal. I have read other reviews and authors' feedback. The rebuttal has addressed most of my concerns. Please add these additional experiments to the final paper/supp. I would keep my initial rating.

---

> > > ### Author Response · Authors · 2023-08-20
> > > **Thank you!**
> > >
> > > Dear Reviewer EXcw,
> > >
> > > Thank you for all feedback and positive comments. We will update our final version accordingly.
> > >
> > > Best,
> > >
> > > The Authors

---

### Official Review · Reviewer_BaoX · 2023-07-07

**Soundness:** 2 fair
**Presentation:** 2 fair
**Contribution:** 2 fair
**Rating:** 5
**Confidence:** 3

**Summary:**

VideoComposer is a tool designed to enhance video synthesis by incorporating textual, spatial, and temporal conditions. It uses motion vectors from compressed videos to guide temporal dynamics and employs a Spatio-Temporal Condition encoder to effectively integrate spatial and temporal relations of inputs. This improves inter-frame consistency and allows for greater control over the synthesized video's spatial and temporal patterns.


**Strengths:**

The VideoComposer offers better control over video synthesis, temporal guidance using motion vectors, improved inter-frame consistency with its Spatio-Temporal Condition encoder, versatility in accepting various forms of inputs, and high customizability, resulting in more precise and desired synthesized videos.

**Weaknesses:**

An ablation study could be conducted on VideoComposer, where each component is removed in turn to evaluate its impact on overall performance. This would help evaluate the value of training under multiple conditions versus a single condition.

Additionally, comparing VideoComposer to a simpler method like Text2Video-Zero [a] with ControlNet [b] would demonstrate whether the increased complexity of VideoComposer yields significantly better results, hence justifying its sophistication.

[a] Text2Video-Zero: Text-to-Image Diffusion Models are Zero-Shot Video Generators, L Khachatryan et al.
[b] Adding Conditional Control to Text-to-Image Diffusion Models, L. Zhang et al.

**Questions:**

How are image-text pairs utilized in the training process of the model?

**Limitations:**

What are the limitations?

---

> ### Author Rebuttal · Authors · 2023-08-09
>
>
> **Q1: An ablation study could be conducted on VideoComposer, where each component is removed in turn to evaluate its impact on overall performance. This would help evaluate the value of training under multiple conditions versus a single condition.**
>
> Thanks for your suggestion. We want to clarify that adding conditions will not necessarily improve the performance, and we do not claim this as a motivation for VideoComposer. Rather, our objective is to enhance the controllability by decomposing videos into various conditions. This approach allows VideoComposer to execute multiple tasks with a single model after just one-time training. Recognizing the validity of the reviewer's suggestion, we aim to validate whether adding more conditions will augment the controllability. Specifically, we seek to better reconstruct the original video by incrementally introducing the textual condition, depth map condition, and single image condition. As illustrated in Figure R8 of the attached one-page PDF file, we observe an improved alignment between the reference video (top row) and the generated videos. If the reviewer raises further concerns, we are more willing to address them and clarify any ambiguities.
>
> **Q2: Additionally, comparing VideoComposer to a simpler method like Text2Video-Zero [a] with ControlNet [b] would demonstrate whether the increased complexity of VideoComposer yields significantly better results, hence justifying its sophistication.**
>
> Thanks for the valuable suggestion. To address this concern, we provide examples to demonstrate the superiority of the depth map-conditioned generation ability of VideoComposer. In Figure R4, we compare our VideoComposer with Text2Video-Zero and existing state-of-the-art Gen-1. We observed that Text2Video-Zero suffers from appearance inconsistency and structural flickering due to the lack of temporal awareness. Meanwhile, Gen-1 produces a video with color inconsistency and structure misalignment (revealed by the orientation of the bird head). The video generated by VideoComposer is faithful to the structure of the input depth sequence and maintains a continuous appearance. This shows the superiority of our VideoComposer in terms of controllability.
>
> **Q3: How are image-text pairs utilized in the training process of the model?**
>
> Thanks. We train VideoComposer jointly on video-text and image-text pairs by treating images as 'one-frame' videos. When using image-text pairs for training, the shape of the input noise is $1 \times h \times w \times c$, where the temporal length is 1 (*i.e.*, $F=1$).
>
> **Q4: What are the limitations?**
>
> Due to the page constraints, we include the discussion of limitations in the Sec. C of the Supplementary Material. In brief, the limitations include **(i)** the use of a watermarked pre-training dataset that leads to visually unappealing videos and **(ii)** the resultant low-resolution videos due to computational constraints.

---

### Official Review · Reviewer_akz1 · 2023-07-07

**Soundness:** 3 good
**Presentation:** 2 fair
**Contribution:** 3 good
**Rating:** 5
**Confidence:** 3

**Summary:**

The paper presents a method for compositional video synthesis. It introduces motion vectors from compressed videos as a control signal for temporal dynamics. The motion vector can be combined by other conditions such as sketch, and depth map. Both qualitative and quantitative results show that the proposed method can control the spatial-temporal patterns.

**Strengths:**

+ The motion controlled generation result (fig 8) using hand-crafted strokes is interesting.

+ Table A1 shows effectiveness of the proposed method quantitatively compared to previous methods.

+ The paper is well written and easy to follow.

**Weaknesses:**

- There are a few GAN-based video synthesis approaches that are worth discussing in the related work. For example, MoCoGAN [1] approaches the problem by decomposing motion and content.

- The two-stage training strategy needs more clarification. What is "compositional training" particularly in the second stage?  How does it differentiate from the "text-to-video" generation in the first stage?

- In line 164-165, the authors "repeat the spatial conditions of a single image and single sketch along the temporal dimension". If the input condition is simply repeated, what's the point of applying a temporal Transformer? It will be equivalent to applying the spatial operation only and repeat at the latent space but with higher computation cost, no? (for motion vector, I totally agree that a spatial-temporal modeling would be necessary.)

- Motion vectors can be less meaningful in the background due to lack of high-level semantics. It can also be clearly seen from the top row in Fig 4. I wonder if the authors treat the motion vector field equally for all locations. It seems that the generated results with motion conditions has more blurry background.

- From Figure 2 and Figure 1(d), my impression is that the conditions (say motion and depth) can be combined together. However, in ablation studies (table 2), only one condition is added at a time. Another ablation that studies all combinations of these conditions will be favored.

[1] Tulyakov, Sergey, et al. "Mocogan: Decomposing motion and content for video generation." CVPR 2018.

**Questions:**

1. In the video translation demo (2:11-2:16), the right example's output does not have a consistent color (white before jumping and brown afterwards). Is there any particular reason why the color consistency fails to hold in such a case?

2. I'd suggest moving some quantitative results (Table A1) to the main text.


**Limitations:**

The authors have addressed the limitations in the supplementary materials.

---

> ### Author Rebuttal · Authors · 2023-08-09
>
> **Q1: Discuss the GAN-based methods like MoCoGAN in the Related Work.**
>
> We greatly appreciate your intention of improving the Related Work by comparing with GANs, such as MoCoGAN.
> - Different motivations. MoCoGAN is an unconditional method that aims to improve the quality of video generation, while VideoCompoer is a conditional approach to enhance the controllability during synthesis.
> - Different methodology. MoCoGAN decomposes videos into content and motion by sampling the latent gaussian noise from different space, but VideoComposer composes a video with textual, spatial, and temporal conditions in the input phase, resulting in a vast design space for customizable video creation.
>
> We will include MoCoGAN and other GAN-based methods in our next version.
>
>
>
> **Q2: Clarifying the two-stage training strategy.**
>
>
> We apologize for unclear presentation. The two-stage training is designed to methodologically address the learning challenge.
>
> - In the first stage, VideoComposer focuses on learning the temporal dynamics by only leveraging the textual condition.  This foundation allows for a focused understanding of temporal relationships within the video content.
> - In the second stage, VideoComposer performs **compositional training** by utilizing textual, spatial, and temporal conditions, building on the temporal modeling ability. The major difference between two stages lies in this incorporation of additional conditions, leading to a comprehensive learning of synthesizing video content from multiple compositions.
>
>
> We have further detailed the explanation of this concept in the revised version.
>
> **Q3: Justifications for "repeat the spatial conditions of a single image and sketch".**
>
> Thanks for pointing out this observation. We greatly value your suggestion to improve the efficiency, and acknowledge that simply repeating the latent feature can be an alternative. Additionally, we also think it to be plausible to utilize the temporal transformer:
> - **Unified interface**: We design the STC-encoder as a unified interface to incorporate different conditions (*i.e.*, single and sequential inputs) without re-designing the architecture, which can equip VideoComposer with better expansibility. Thus, repeating the input can adhere to this spirit well.
> - **Minimal computational cost**: The computational cost of the temporal transformer is negligible compared to the 3D UNet. Specifically, the computational overhead for inferring one STC-encoder is just 0.042\% of that required by the 3D UNet.
>
>
>
>
> **Q4: Question about the utilization of motion vectors and the resultant blurry background in Figure 4.**
>
> Thank you for the careful questions. This question touches on two aspects:
>
>
> - Treatment of Motion Vectors. In Figure 4, we treat the motion vector field equally for all locations. However, we offer flexibility during inference by using partial motion vectors.
> - Background Blurriness in Figure 4. We attribute it to the motion magnitude difference between two examples. The first video in the tiger example of Figure 4 contains minimal magnitude of motion, thus generating clear background. In comparison, the second video in the tiger example of Figure 4 (*i.e.*, the example in Figure R1(b)) utilizes motion vectors displaying large magnitude of motion, thereby easily resulting in blurriness. If we constrain the motion to a small magnitude as shown in Figure R1(a), we obtain clearer background.
>
>
>
>
> **Q5: More ablation studies of combining more conditions should be provided in Table 2.**
>
> Thanks for raising this valuable concern.
>
> - **The purpose of Table 2**: As the reviewer mentioned, multiple conditions can be utilized together to guide the generated videos. However, we want to clarify that the aim of Table 2 is to demonstrate the usefulness of the STC-encoder in enhancing the temporal awareness of input conditions. By comparing results with and without STC-encoder with identical input conditions, we can highlight such effectiveness, using the metric of frame consistency. We do not expect that adding more conditions will consistently improve this metric.
> - **More comprehensive study**: Recognizing the validity of your suggestion, we have expanded Table 2 to encompass more conditions. Additional results presented in Table R1 demonstrate how the STC-encoder functions effectively with various combinations of conditions, confirming its versatility and applicability.
>
> **Table R1**: Quantitative ablation study of STC-encoder. "Conditions" denotes the conditions utilized for generation.
>
>
>
> |Methods|Conditions|Frame consistency|
> |:-|:-:|:-:|
> |*w/o* STC-encoder / VideoComposer|Text, sketch sequence, and depth sequence|0.911 / **0.918**|
> |*w/o* STC-encoder / VideoComposer|Text, sketch sequence, and motion vectors|0.912 / **0.919**|
> |*w/o* STC-encoder / VideoComposer|Text, depth sequence, and motion vectors|0.916 / **0.923**|
> |*w/o* STC-encoder / VideoComposer|Text, sketch sequence, depth sequence, and motion vectors|0.914 / **0.920**|
>
> **Q6: Color inconsistency of the tiger in video translation demo (2:11-2:16).**
>
>
> Thank you for the careful observations. The problem of color consistency is a long-standing challenge in video generation. In this particular example, we conjecture that this stems from **the lack of an explicit textual condition**: The web-scale pre-training data contains both white and brown tigers. The coarse description applied doesn't specifically differentiate between them. We also provide **one possible solution** to address this inconsistency: providing more specified textual instructions, particularly regarding the color of the tiger. By regenerating the video using an identical random seed, as shown in Figure R2 (below), we achieve more satisfactory color consistency in the resultant video.
>
>
>
> **Q7: Moving results in Tab. A1 to the main paper.**
>
> Thanks for the valuable suggestion. We will move it to the main paper to ensure a comprehensive comparison.

---

> > ### Comment · Reviewer_akz1 · 2023-08-16
> >
> > I would like to thank the authors for responding to my questions. My concerns have been well addressed and I believe adding these into the revision would strengthen the paper. Therefore I would like to lift up my rating.

---

> > > ### Author Response · Authors · 2023-08-17
> > > **Thank you!**
> > >
> > > Dear Reviewer akz1,
> > >
> > > Thanks for raising the score rating. We appreciate your efforts in the reviewing process and all useful feedback to improve our manuscript. We will update our final manuscript to reflect all the modifications.
> > >
> > > Best,
> > >
> > > The Authors

---

### Author Rebuttal · Authors · 2023-08-09

We appreciate the reviewers for their positive comments and constructive feedback on our paper. We are encouraged that VideoComposer is recognized for its merits, including

- clarity in presentation [Reviewer akz1, EXcw]
- superior performance both quantitatively and qualitatively over prior methods [Reviewer akz1, Pp5z]
- introduction of innovative motion vectors [Reviewer Pp5z, BaoX]
- a novel interface for handling inputs [Reviewer Pp5z, BaoX]
- its heightened controllability and customizability [Reviewer akz1, BaoX, EXcw].

We also acknowledge the concerns raised and will address them comprehensively. These insightful reviews lead to multiple improvements of our original manuscript.

In this rebuttal, we have included some new figures in **the newly uploaded one-page PDF**. Here, we briefly summarize the content of these figures to facilitate quick reference.

- Figure R1 shows the effect of varying motion magnitude by sampling at different frame rates of the motion vector condition. [Reviewer akz1]
- Figure R2 gives the video translation results, suggesting that we can leverage a detailed prompt to generate a more desired and color-consistent video. [Reviewer akz1]
- Figure R3 gives the compositional video generation results, where the video is generated using text, image and mask sequence as conditions. [Reviewer EXcw]
- Figure R4 compares VideoComposer with other existing methods such as Text2Video-Zero and Gen-1. [Reviewer BaoX, Pp5z]
- Figure R5 shows the comparison of VideoComposer using STC-encoder with and without weight sharing on video inpainting task. [Reviewer EXcw]
- Figure R6 includes the experiment of motion control with single sketch condition. [Reviewer EXcw]
- Figure R7 compares VideoComposer with VideoP2P and vid2vid-zero.  [Reviewer EXcw]
- Figure R8 shows the comparison of video generation with single and multiple conditions. [Reviewer BaoX]


We are open to discussions and are committed to addressing any concerns from the reviewers, ensuring the continual refinement of VideoComposer.

---

### Decision · Program_Chairs · 2023-09-21

**Decision:**

Accept (poster)

**Comment:**

This paper receives consistent review scores of 5/6/5/6.

The paper presents VideoComposer, a model which allows users to flexibly compose a video with textual, spatial, and temporal conditions. Both qualitative and quantitative results show that the proposed method can synthesize controllable high-quality videos.

All reviewers appreciate the contributions of the paper in controllable video synthesis. Almost all concerns they raised in the first round of review have been addressed in the rebuttal.

The authors are suggested to include the additional ablation studies and clarifications they provided during rebuttal in the final version.